

# Distributed Bayesian networks reconstruction on the whole genome scale

Alina Frolova[1] and Bartek Wilczyński[2]

[1] Institute of Molecular Biology and Genetics, Kyiv, Ukraine
[2] Institute of Informatics, University of Warsaw, Warsaw, Poland

## ABSTRACT

**Background**. Bayesian networks are directed acyclic graphical models widely used to represent the probabilistic relationships between random variables. They have been applied in various biological contexts, including gene regulatory networks and protein–protein interactions inference. Generally, learning Bayesian networks from experimental data is NP-hard, leading to widespread use of heuristic search methods giving suboptimal results. However, in cases when the acyclicity of the graph can be externally ensured, it is possible to find the optimal network in polynomial time. While our previously developed tool BNFinder implements polynomial time algorithm, reconstructing networks with the large amount of experimental data still leads to computations on single CPU growing exceedingly.

**Results**. In the present paper we propose parallelized algorithm designed for multi-core and distributed systems and its implementation in the improved version of BNFinder— tool for learning optimal Bayesian networks. The new algorithm has been tested on different simulated and experimental datasets showing that it has much better efficiency of parallelization than the previous version. BNFinder gives comparable results in terms of accuracy with respect to current state-of-the-art inference methods, giving significant advantage in cases when external information such as regulators list or prior edge probability can be introduced, particularly for datasets with static gene expression observations.

**Conclusions**. We show that the new method can be used to reconstruct networks in the size range of thousands of genes making it practically applicable to whole genome datasets of prokaryotic systems and large components of eukaryotic genomes. Our benchmarking results on realistic datasets indicate that the tool should be useful to a wide audience of researchers interested in discovering dependencies in their large-scale transcriptomic datasets.

Corresponding author
Alina Frolova, fshodan@gmail.com, a.o.frolova@imbg.org.ua

## INTRODUCTION

Bayesian networks (BNs) are graphical representations of multivariate joint probability distributions factorized consistently with the dependency structure among variables. In practice, this often gives concise structures that are easy to interpret even for non-specialists.

A BN is a directed acyclic graph with nodes representing random variables and edges representing conditional dependencies between the nodes. Nodes that are not connected represent variables that are independent conditionally on their parent variables (*Friedman & Koller, 2003*). In general, inferring BN structure is NP-hard (*Chickering, Heckerman & Meek, 2004*), however it was shown by *Dojer (2006)* that it is possible to find the optimal network structure in polynomial time when datasets are fixed in size and the acyclicity of the graph is pre-determined by external constraints. The acyclicity can be ensured in two cases: when inferring dynamic BNs from time-series data (the "unrolled" graph, i.e., graph with a copy of each variable in each discretized time point, is always acyclic), or when user defines the regulation hierarchy restricting the set of possible edges in case of static BNs (for example, by introducing the list of potential regulators). Therefore the minimal information required to infer dynamic BN is expression values matrix, and in case of static BN—expression values matrix and the list of potential parent variables.

This algorithm was implemented in BNFinder—a tool for BNs reconstruction from experimental data (*Wilczyński & Dojer, 2009*).

One of the common use of BNs in bioinformatics is inference of interactions between genes (*Zou & Conzen, 2005*) and proteins (*Jansen et al., 2003*). Even though it was originally developed for this purpose, BNFinder is a generic tool for reconstructing regulatory interactions. Since its original publication, it was successfully applied to linking expression data with sequence motif information (*Dabrowski et al., 2010*), identifying histone modifications connected to enhancer activity (*Bonn et al., 2012*) and to predicting gene expression profiles of tissue-specific genes (*Wilczynski et al., 2012*).

BNFinder is implemented in Python programming language and is therefore available for most modern operating systems. It is a command line tool, but the usage is quite easy even for the scientists without a strong Computer Science background. The learning data must be passed to BNFinder in a text file split into 3 parts: preamble, experiment specification and experiment data. The preamble allows users to specify some features of the data and/or network, while the next two parts contain the learning data, essentially formatted as a table with space- or tab-separated values. For example, if the user wants to infer the network with four genes {G1, G2, G3, G4}, where G1 and G2 are known transcription factors, the input file would look like Table 1.

In cases when more information is available about the experiment, the preamble section of the input file can be used to incorporate perturbational data, prior probabilities of the genes interaction or the expected structure of the signaling pathway (especially in cases when one expects cascade activation of the regulatory factors). These parameters are not required but they can substantially increase the accuracy of inferred network. Readers interested in applying BNFinder to their own data might find it useful to look through the BNFinder tutorial (https://github.com/sysbio-vo/bnfinder/raw/master/doc/bnfinder_tutorial.pdf), which includes several working examples of simple and more complex usage scenarios.

BNFinder supports several output formats, the simplest of which is SIF (Simple Interaction File), where each line represents the fact of a single interaction between two variables. The SIF output for the network in Table 1 shows positive correlation between

**Table 1** **An example of BNFinder input learning data for inferring static BN with four genes, where genes G1 and G2 are regulators (transcription factors).** The first line denotes preamble (in this case - regulators list), the second line—experiment specifications (the list of experiments names), and the rest of lines denote experiment data (gene expression values).

**#regulators G1 G2**

| conditions | EXP0 | EXP1 | EXP2 | EXP3 | EXP4 | EXP5 |
|---|---|---|---|---|---|---|
| G1 | 0.2 | 3.4 | 1.3 | 7.4 | 2.2 | 0.4 |
| G2 | 4.5 | 7.8 | 0.3 | 5.6 | 3.3 | 1.1 |
| G3 | 1.0 | 2.9 | 0.8 | 5.5 | 1.6 | 2.8 |
| G4 | 3.2 | 6.5 | 0.5 | 3.1 | 8.2 | 5.0 |

G1 and G3, G2 and G4 genes:

```
G1 + G3
G2 + G4
```

The detailed explanation on all input parameters and their influence on output data is described in the user manual, freely available from the dedicated github repository— https://github.com/sysbio-vo/bnfinder/raw/master/doc/bnfinder_doc_all.pdf.

Even though BNFinder can be applied to many different datasets, the practical usage of the algorithm is limited by its running times that can be relatively long. Since the algorithm published by *Dojer (2006)* has the capacity to be parallelized by design and the current version of BNFinder (*Dojer et al., 2013*) has only a simple parallelization implemented, we have developed a new version that takes advantage of multiple cores via the python multiprocessing module and gives better performance.

## IMPLEMENTATION

The general scheme of the learning algorithm is the following: for each of the random variables find the best possible set of parent variables by considering them in a carefully chosen order of increasing scoring function. This function consists of two components: one is penalizing the complexity of a network and the other one is evaluating the possibility of explaining data by a network. The acyclicity of the graph allows to compute the parents set of each variable independently, therefore algorithm first starts with computing scores for all possible singleton parents sets for a given random variable. Then it checks if penalty on increasing parents set size is too high or if it reached user defined parents set size limit, and if no—proceeds with two-element parents sets and so on. The detailed description of the algorithm and scoring functions is given in Dojer manuscript (*Dojer, 2006*).

Current parallelization in BNFinder version 2 (*Dojer et al., 2013*) can be considered **variable-wise** as it distributes the work done on each variable between the different threads. However, such approach has natural limitations. Firstly, the number of parallelized tasks cannot exceed the number of random variables in the problem, meaning that in the cases where only a few variables are considered (e.g., in classification by BNs) we get a very limited performance boost. Secondly, **variable-wise** parallelization might be vulnerable (in terms of performance) to the datasets with highly heterogeneous variables, i.e., variables

whose true dependency graph has a wide range of connections. As the time spent on computing parent sets for different variables varies - it leads to uneven load of threads. In biology we usually observe networks with scale-free topology consisting of a few hub nodes with many parents and a large number of nodes that have one or small number of connections (*Barabasi & Oltvai, 2004*). If one applies **variable-wise** algorithm to such networks the potential gain in the algorithm performance is not greater than in the case where all the nodes have as many parents as the hub node with the largest parent set.

While **variable-wise** algorithm is the most straightforward one, it is also possible to consider different possible parents sets in parallel, as is the case for the **set-wise** algorithm. It means that in the first step we compute singleton parents sets using all available threads, in the second step we compute two-element parents sets in parallel and so on, until we reach parents sets size limit or scoring function limit. However, the **set-wise** algorithm requires more synchronizations between the threads (*McCool, Reinders & Robison, 2012*) in comparison with the **variable-wise**. On top of that allocating large number of cores to the variable whose parents set is very quick to compute may result in lower performance due to the context switching. The latter is the process of storing the state of a thread, so it can be restored and execution resumed from the same point later. Context switch is governed by operating system and has a cost in performance, therefore if we allocate 100 cores to the variable which true singleton parent is found in the first step (which is quick to compute as we only need to score single parent-variable interactions), it would probably take more time than when allocating 10 cores. As it is difficult to tell, which problem might be more important in practice, we have implemented and tested two approaches: **set-wise** only and **hybrid** one - a natural combination of **variable-wise** and **set-wise**. It should be noted that, while the variable-wise parallelization was already implemented in the BNFinder, the set-wise and hybrid methods are novel in this particular application.

Figure 1 shows parallelization algorithms schema in a simplified way for better understanding. As stated above, **set-wise** algorithm uses each given core to compute parents sets for one gene and after finding parents it proceeds with the next gene. On the contrary, **hybrid** algorithm uniformly distributes cores between genes, for example, if a user has three genes in the network and six cores available, each gene will have two cores for computing its parents set. If there are seven cores available, one gene will have three cores, while other two genes will have two cores. Thus, once the gene is processed, the freed cores cannot be allocated to other genes, which may be a potential disadvantage.

So, the pure theoretical complexity of **set-wise** (left side of inequality) and **hybrid** (right side of inequality) algorithms can be described in the following way:

$$\frac{\sum_{i=1}^{n} t_i}{k} = \frac{(\text{avg}_{i=1}^{n} t_i)n}{k} \leq \frac{(\text{max}_{i=1}^{n} t_i)n}{k}$$

where $k$ is the cores number, $n$ is the number of random variables, and $t_i$ is the time one needs to compute optimal parents set for the $i$th variable using one core.

Thus, the time to reconstruct the whole network in case of a **set-wise** approach is the sum of time needed for each random variable, which is in fact average time one spends on finding the parents set for one variable, while inferring BN with a **hybrid** approach is

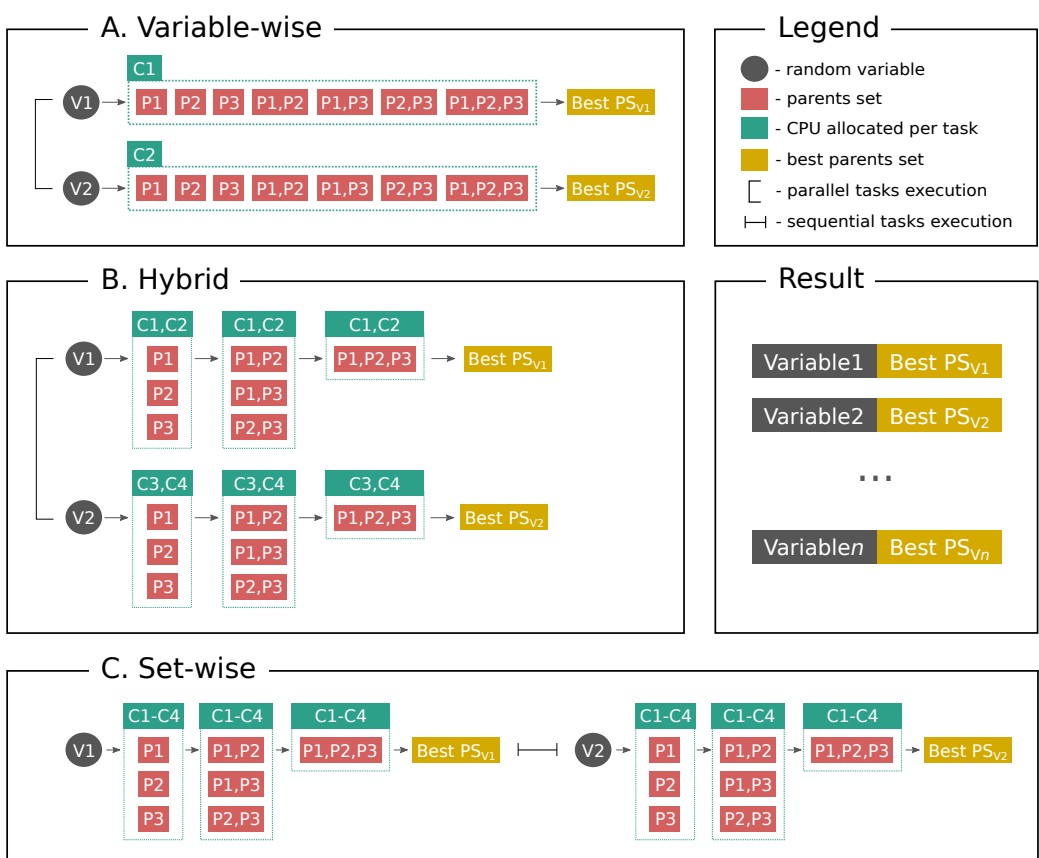

**Figure 1 Parallelization algorithms schema.** For simplicity the plot depicts the case with two random variables (genes, proteins or other biological entities), three parents (regulators), and four CPUs. However, this example can be easily extended to more variables, and in the end the user gets output as shown on **Result** subplot. After considering parents sets of a same size BNFinder checks if either parents set size limit is reached or scoring function penalty is too high to decide if proceed next. (A) **Variable-wise** algorithm can only parallelize between variables, therefore out of four CPU it uses only two. (B) **Hybrid** algorithm uses two levels of parallelization—between variables and between parents sets of each variable, thus, it uses two CPUs per variable in a given example. In general it uniformly distributes cores among variables. (C) **Set-wise** algorithm implements parallelization between parents sets only, thus, considering variables sequentially and using four CPUs per variable.

bounded by the maximum time one spends on one variable. This estimate does not take into account the time, which operating system spends on multi-threading or accessing the disk storage.

# RESULTS

## Performance testing
### *Algorithms comparison*

We compared implementations of three different algorithms: **variable-wise**, **set-wise** and **hybrid**. The original implementation (**variable-wise**) serves as a baseline for computing the speed-up and efficiency of the parallelization. For testing we used synthetic benchmark
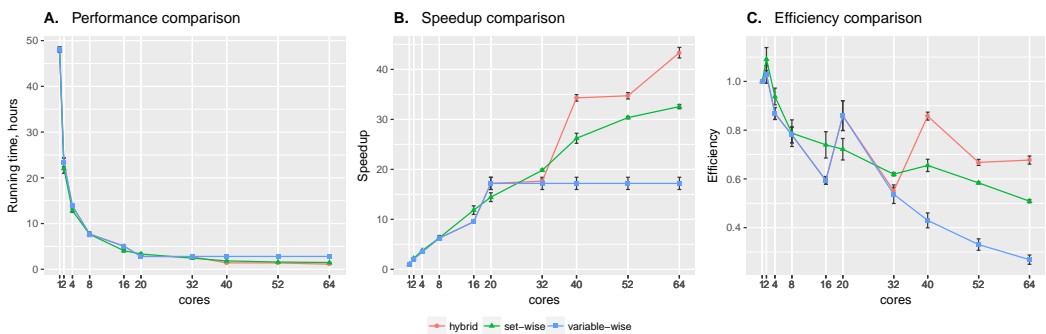

**Figure 2** **Synthetic data testing.** Comparing performance (A), speed-up (B) and efficiency (C) of algorithms on synthetic benchmark data: 20 variables ×2,000 observations. Speed-up is a ratio of the performance with increase of the cores number. Efficiency is a speed-up normalized by the number of cores.

data as well as real datasets concerning protein phosphorylation network published by *Sachs et al. (2005)*. The efficiency is defined as speed-up divided by the number of cores used.

**Set-wise** and **hybrid** algorithms performance on a 20-gene synthetic network was very similar, while the speed-up and efficiency comparison revealed more differences between the algorithms (see Fig. 2). There is no regulators list for this network, so BNFinder reconstructed a dynamic BN. **Hybrid** algorithm showed more unstable behavior, performing better when the number of cores correlates with the number of genes. It outperformed **set-wise** when the number of cores exceeded the number of genes two times at least, however it didn't show any speed-up when increasing the number of cores from 42 to 50. The latter is easily explained by the algorithm design, since running time is bound by the most computationally complex variable, using 41–59 cores cannot give performance boost as long as this variable provided with one core only.

The Sachs et al. network we tried next has 11 proteins. The regulators are selected from those proteins and introduced in the cascade manner, which denotes expected layer structure of the signaling pathway. The derivation of the structure relied on the simultaneous measurement of multiple phosphorylated protein and phospholipid components in thousands of individual primary human immune system cells. Perturbing these cells with molecular interventions drove the ordering of connections between pathway components. We took the data from the article, and transformed it into the format suitable for BNFinder. In particular, first layer can regulate all the following, while each next one cannot regulate previous layers. On the first layer only singleton parents set is possible consisting of *plcg* protein, on the second layer we have two regulators, *PIP3* and previously defined *plcg*, making it possible to search for singleton and two-element parents sets for the rest of proteins, and so on.

Sachs et al. data showed significant difference between two algorithms. Clearly, the **set-wise** algorithm outperforms the **hybrid** one: using 11 cores it showed 8× speed-up, while the **hybrid** algorithm showed only 1.5× speed-up (see Fig. 3). **Hybrid** algorithm

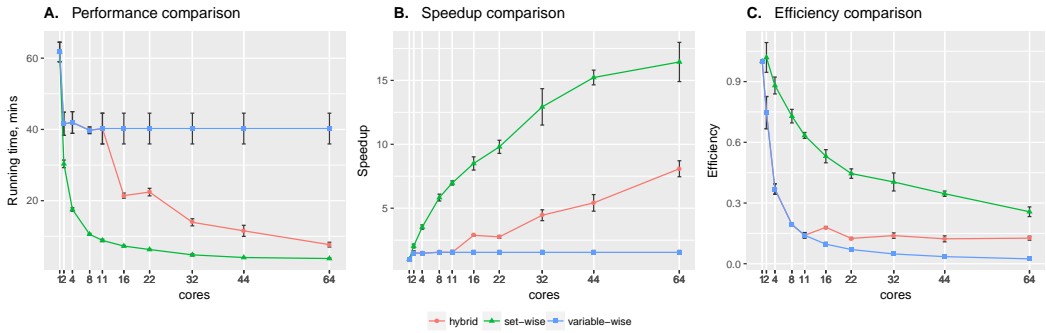

**Figure 3 Sachs et al. data testing.** Comparing performance (A), speed-up (B) and efficiency (C) of algorithms on protein phosphorylation data: 11 variables × 1,023 observations (*Sachs et al., 2005*). Speed-up is a ratio of the performance with increase of the cores number. Efficiency is a speed-up normalized by the number of cores.

performance was hindered by highly heterogeneous variables in the input data, because out of the 11 proteins in the network *pakts473* has 6 parents, *p44/42* has 3 parents, while others have 1–2 parents. Importantly, the better performing algorithm is also the one showing more consistent behavior.

However, the way the regulators are introduced in the input file produces uneven load by itself, as each next variable has bigger set of potential parents. Therefore, we generated more benchmark data with different number of regulator and target genes, where we could define regulators in one layer manner or similar to Sachs data. Moreover, the underlying structure of generated networks was designed to be heterogeneous. Namely, it contains genes with gradually increasing number of parents: first gene has zero regulators, second gene—one regulator, third—two regulators and so on. The datasets were generated with *BayesGen.py* script that is included in the supplementary material. It takes the desired connectivity between variables and simulates the observations as emissions from a BN with bimodal Gaussian distributions of variables.

The results of multiple tests showed that introducing complex layer structure of regulators always resulted in the **hybrid** algorithm poor performance. It either showed much worse results regardless of the number of cores as in Fig. 3 or it showed comparable speed-up when the number of cores was three times bigger than the number of genes. In cases when regulators were supplied as one single list both algorithms showed results similar to those in Fig. 2, namely the **set-wise** algorithm was better when the number of cores was lower than the number of genes, while the **hybrid** algorithm was better in the opposite case (although there was no such dramatic difference as in the case of the layered regulators structure). The more observations per regulator-target interaction we had, the better BNFinder predicted the network structure. However, as we studied running times per gene we observed that variables with the biggest number of parents did not always result in the longest computations. The latter is explained by how the scoring function works. BNFinder stops when the penalty for increasing the set of parents is so big that it cannot improve beyond what it has already found. In general, if the optimal parent set

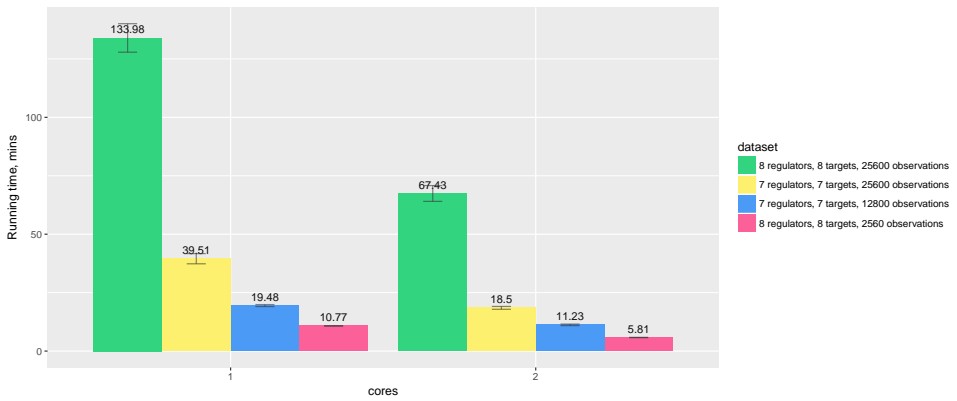

**Figure 4** **Hybrid algorithm performance on the datasets of different size.** The tendency is preserved for larger numbers of cores as well.

is very good in predicting the child variable value BNFinder will finish searching earlier. It means that the whole family of three-element parents sets can have worse scores than the two-element parents set, but the algorithm will proceed further because it has not yet reached the penalty on increasing the set size.

In particular, running times also depend on the number of observations and the number of nodes in the networks. For example, Fig. 4 shows that increasing the number of observations by $10\times$ leads to $12\times$ longer running time for the network of the same size, while 8-gene networks take 3 times longer to compute in comparison to 7-gene networks having the same number of observations.

Since there is no obvious winner between **set-wise** and **hybrid** algorithms, we decided to provide users with both options with **set-wise** being the default algorithm.

Tests on benchmark and Sachs data were performed on the same server with AMD Opteron (TM) Processor 6,272 (4 CPUs with total of 64 cores) and 512GB RAM. During the tests the server was loaded only by regular system processes, but to ensure statistical significance we performed each test five times, plotting average values with standard deviations.

### Distributed computations testing

For the new version of BNFinder we also implemented an option for distributed usage of the tool. The idea is quite simple and did not require specific python libraries or tools. The user has to submit the file with subset of genes as input argument, so BNFinder can calculate partial result. When all the genes are processed user must place all the results into one folder and run BNFinder again, so it will aggregate the results.

For the tests we chose Challenge 5 (Genome-Scale Network Inference) data from DREAM2 competition (Dialogue for Reverse Engineering Assessments and Methods) (*Stolovitzky, Prill & Califano, 2009*; *DREAM Initiative, 2009*). Challenge data is a log-normalized compendium of Escherichia coli expression profiles, which was provided by Tim Gardner to DREAM initiative (*Faith et al., 2007*). The participants were not informed

**Table 2 DREAM2 Challenge 5 data testing.** BNFinder **set-wise** algorithm is used with 4 cores and 4 genes per task. The tasks were then distributed among computational nodes. *l* stands for BNFinder parents sets limit. CLR with cutoff means limiting output results to 100,000 genes interactions.

| | CLR | CLR with cutoff | BNF, $l = 1$ | BNF, $l = 2$ | BNF, $l = 3$ Intel Xeon E5345 | BNF, $l = 3$ Intel Xeon E5-2690 |
|---|---|---|---|---|---|---|
| CPU time, hours | 0.7879 | 0.1999 | 2.0021 | 383.7149 | 109,200 | 54,800 |
| Actual time, hours | 0.2626 | 0.0666 | 0.0667 | 12.7904 | 336 | 169 |
| Total CPU number | 3 | 3 | 30 | 30 | ∼325 | ∼325 |

about the data origin and were provided only with 3,456 genes × 300 experiments dataset and the list of transcription factors.

BNFinder was tested with different parents set limit parameter value (i.e., maximum number of potential parents), which increases the computation time dramatically in non-linear way, especially in case of dataset with many variables. We compared **set-wise** algorithm performance with context likelihood of relatedness (CLR) algorithm - an extension of the relevance networks approach, that utilizes the concept of mutual information (*Faith et al., 2007*). We chose CLR, because it is very fast and easy to use tool, which provides good results. In addition, CLR-based algorithm - synergy augmented CLR (SA-CLR) was best performed algorithm on Challenge 5 (*Watkinson et al., 2009*).

The CLR tests were performed on the GP-DREAM platform, designed for the application and development of network inference and consensus methods (*Reich et al., 2006*). BNFinder tests were done within Ukrainian Grid Infrastructure (*VO "Infrastructure", 2011*), which was accessed through nordugrid-arc middleware (arc client version 4.1.0 (*Ellert et al., 2007*)), so the tasks submitting process was automated and unified. The results presented in Table 2 are not precisely comparable due to differences in used hardware, especially when using such heterogeneous environment as Grid. In addition, we could not obtain stable number of cores over time with the Grid, as the clusters were loaded with other tasks. However, running times can give a rough estimate for those who plan to use BNFinder on large datasets.

Even though computing with parents sets limit of 3 takes significant amount of time and resources, it is clear that BNFinder is able to reconstruct genome-scale datasets, significantly broadening its application range after it was adapted to parallel and distributed computing.

## Accuracy testing

Previously we compared accuracy of BNFinder algorithm with Banjo (*Wilczyński & Dojer, 2009*) on data provided with the tool and separately on Sachs data (*Dojer et al., 2013*), which we used in this work to test the performance. Here we performed accuracy testing on 14 different datasets, both synthetic and taken from microarray experiments. No other additional data or information was used for the networks inference except those mentioned in the datasets description below.

**DREAM2 Genome Scale Network Inference** 3,456 genes ×300 experiments dataset, log-normalized compendium of Escherichia coli expression profiles described above (*Stolovitzky, Prill & Califano, 2009*). Transcription factors list is provided with the data.

**Table 3** Gene regulatory network inference methods used for benchmarking.

| Inference Approach | Method |
|---|---|
| Co-expression algorithms | MutRank (*Obayashi & Kinoshita 2009*) |
| Information-theoretic approaches | CLR (*Faith et al. 2007*), ARACNE (*Margolin et al. 2006*), PCIT (*Reverter & Chan 2008*), C3NET (*Altay & Emmert-Streib 2010*) |
| Feature selection approaches | MRNET (*Meyer et al. 2007*), MRNETB (*Meyer et al. 2010*), Genie3 (*Huynh-Thu et al. 2010*) |
| Bayesian model averaging | FastBMA (*Hung et al. 2017*) |

**DREAM4 In Silico Network Challenge:** time course datasets showing how the simulated network responds to a perturbation and how it relaxes upon its removal. There are 5 different datasets with 10 and 100 genes each. For networks of size 10, datasets consist of 5 different time series replicates, while networks of size 100 has 10 time series replicates. Each time series has 21 time points (*Stolovitzky, Monroe & Califano, 2007*).

**Yeast time series:** 102 genes $\times$ 582 experiments datasets with time series after drug perturbation from the yeast rapamycin experiment described in *Yeung et al. (2011)*. There are $582/6 = 97$ replicates (the 95 segregants plus two parental strains of the segregants), each with measurements at 6 time points. Prior probabilities of gene regulations are provided.

**Yeast static:** 85 genes $\times$ 111 experiments subset of the data used for network inference in yeast by *Brem & Kruglyak (2005)*. Prior probabilities of genes regulations are provided. TF-gene regulations were extracted from YEASTRACT repository (http://www.yeastract.com, version 2013927).

**Yeast static synthetic:** 2,000 genes $\times$ 2,000 experiments dataset generated using GNW simulator (*Schaffter, Marbach & Floreano, 2011*) by extracting parts of known real network structures capturing several of their important structural properties. To produce gene expression data, the simulator relies on a system of non-linear ordinary differential equations. TF-gene regulations were extracted from YEASTRACT repository (version 2013927). The adjacency matrix of true underlying network structure of this dataset has symmetrical form, therefore it is not possible to evaluate the direction of interaction in this case.

DREAM2 data was downloaded from the challenge website, DREAM4, YeastTS (time series), and Brem data was imported from NetworkBMA (*Young, Raftery & Yeung, 2014*) R package, while synthetic Yeast static data (GNW2000) was imported from NetBenchmark (*Bellot et al., 2015*) R package.

Table 3 summarizes gene regulatory network inference methods we used for benchmarking. Specifically, we used FastBMA method implemented in NetworkBMA R package, while the rest of the methods were accessed through NetBenchmark, a bioconductor package for reproducible benchmarks of gene regulatory network inference. The methods were used with default parameters, while for FastBMA, Genie3 and BNFinder prior edge probabilities and TF-gene regulations were supplied where applicable.

We used two metrics to assess methods performance: Area Under the Precision Recall curve (AUPR) or Area Under Receiver Operating Characteristic curve (AUROC), implemented in MINET R package (*Meyer, Lafitte & Bontempi, 2008*). However, this
**Table 4  DREAM2 Genome Scale Network Inference.** BNFinder and CLR are compared with the best scored method using 100% of output interactions. BNFinder is used with parents sets limit 1 and suboptimal parents sets 100, CLR is used with default parameters. Evaluation of BNF on TF-TF free gold standard is provided as well. All the output interactions are considered for calculating areas under curves.

|       | CLR      | BNF      | BNF, gold standard without TF-TF | Winner: Team 48 |
|-------|----------|----------|----------------------------------|-----------------|
| AUPR  | 0.051398 | 0.028769 | 0.030584                         | 0.059499        |
| AUROC | 0.617187 | 0.606326 | 0.629420                         | 0.610643        |

approach gives an estimation of the global behavior of the method, therefore in NetBenchmark package *Bellot et al. (2015)* evaluated the inferred networks using only the top best 20% of the total number of possible connections. The latter allows us to correctly compare methods with sparse and concise outputs. We used both MINET and NetBenchmark evaluation functions in order to assess the impact on methods rankings.

FastBMA, Genie3 and BNFinder methods allow us to infer directed interactions, therefore they were additionally evaluated on directed gold networks (where applicable), while for the undirected evaluation gold network adjacency matrices were converted to symmetrical ones (higher edge probabilities are preserved) as well as outputs of directed methods.

In case of static gene expression data FastBMA can infer the regulators of a particular gene by regressing it on the expression levels of the other genes. Therefore we used the method with time series data only.

We mostly used DREAM2 data for running times tests and evaluated accuracy of CLR method only. The results of BNFinder and CLR tests are shown in Table 4.

We ranked AUROC and AUPR values across all the methods for each of five 10 and 100 genes network from DREAM4 challenge. Different evaluation strategies for 10-gene networks showed quite a different results (Figs. 5 and 6), while 100-gene networks results were more consistent among MINET and NetBechmark (Figs. 7 and 8). We believe that networks of a small size might not be a good benchmark data as even a slightest change in the obtained scores might disrupt rankings dramatically. This is especially valid for the methods, which output might differ each run due to the nature of underlying algorithms (e.g., regression, greedy hill climbing). There is no single best method for DREAM4 data, while Genie3, MRNET, MRNETB, FastBMA, and BNFinder scored first at least once.

Yeast time series network inference showed extremely bad results for all the methods with MutRank having slightly better AUROC = 0.58 and AUPR = 0.07 values according to MINET package. In contrast to synthetic DREAM4 data which has 21 time points, YeastTS has only 6, which could explain worse results.

Surprisingly, BNFinder significantly outperformed other methods when reconstructing network from Brem at al. static gene expression data (Fig. 9). Importantly, Genie3 method was also supplied with the same regulators list as BNFinder, but it has led to worse results contrary to using Genie3 without regulators. The other methods were used without any additional prior information as implemented in NetBenchmark package.

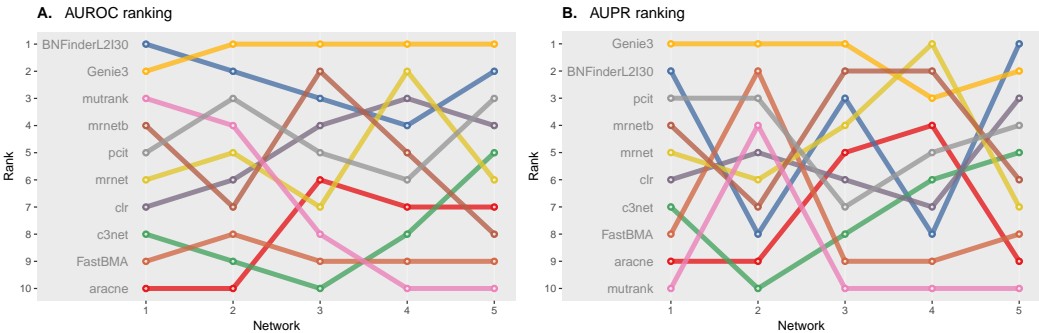

**Figure 5** **DREAM4 10 genes network, evaluation by MINET package.** Area under ROC (A) and PR (B) curves are ranked across different methods: the highest value is ranked first, and the lowest—ranked last. BNFinder is used with parents sets limit 2 and suboptimal parents sets 30.

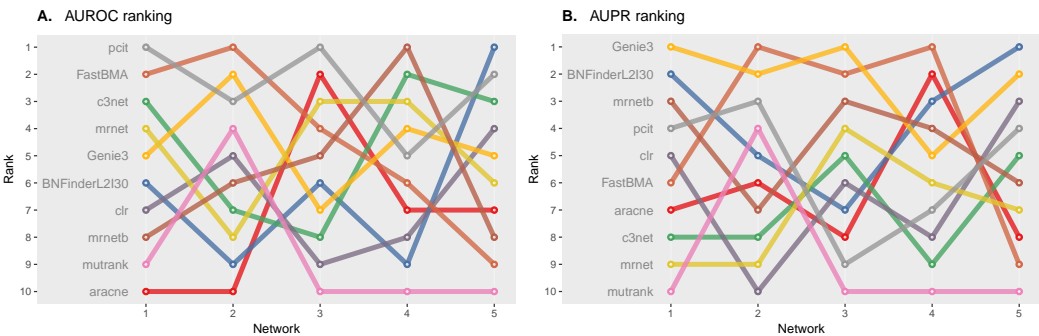

**Figure 6** **DREAM4 10 genes network, evaluation by NetBenchmark package.** Area under ROC (A) and PR (B) curves are ranked across different methods: the highest value is ranked first, and the lowest— ranked last. BNFinder is used with parents sets limit 2 and suboptimal parents sets 30.

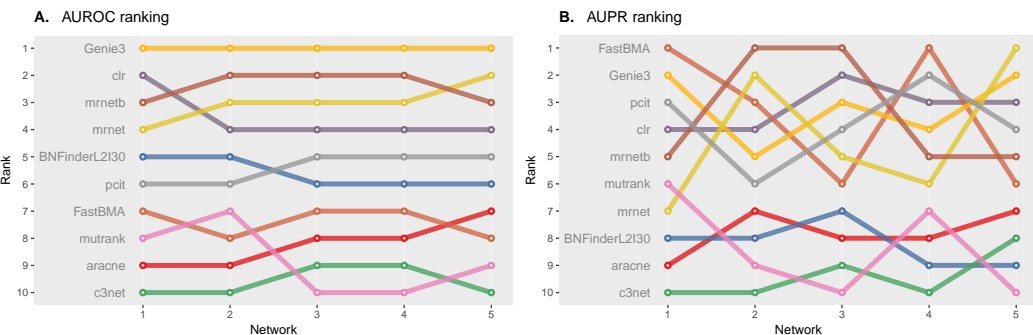

**Figure 7** **DREAM4 100 genes network, evaluation by MINET package.** Area under ROC (A) and PR (B) curves are ranked across different methods: the highest value is ranked first, and the lowest—ranked last. BNFinder is used with parents sets limit 2 and suboptimal parents sets 30.

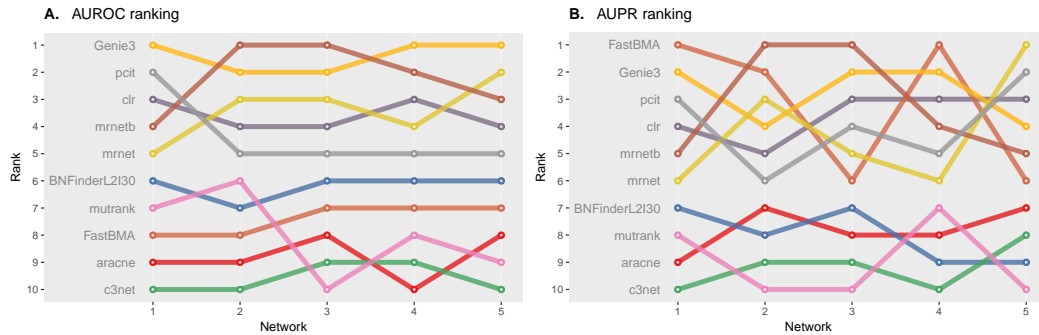

**Figure 8** **DREAM4 100 genes network, evaluation by NetBenchmark package.** Area under ROC (A) and PR (B) curves are ranked across different methods: the highest value is ranked first, and the lowest—ranked last. BNFinder is used with parents sets limit 2 and suboptimal parents sets 30.

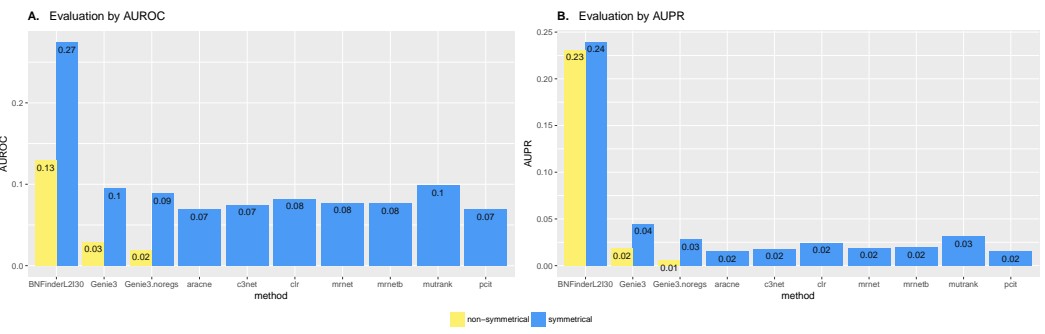

**Figure 9** **Brem et al. Yeast dataset, evaluation by NetBenchmark package.** Directed and non-directed (with symmetrical adjacency matrix) gold standard network evaluation is shown in different colours. Area under ROC (A) and PR (B) curves values are considered. Genie3.noregs is the result of Genie3 execution without the regulators list. BNFinder is used with parents sets limit 2 and suboptimal parents sets 30.

We also studied the effect of the number of experiments on the accuracy of inferred network. For GNW2000 synthetic Yeast data we performed two separate tests: one with full dataset—2,000 experiments, and second with only 150 randomly selected observational points. Figure 10 clearly shows that all the methods improved their results on the full dataset, with BNFinder being among the top methods and having best AUPR on the reduced dataset. Interestingly, we did not see such a major difference between BNFinder and other methods in contrast to Brem at al. data, given the same input parameters and both datasets being of Yeast origin. It shows the importance of developing new gold standards based on experimental data from model organisms as synthetic data only cannot reflect all the complexity of biological interactions.

*Exploring BNFinder parameter space.* The main advantage of BNFinder in comparison with heuristic search Bayesian tools such as Banjo is that BNFinder reconstructs optimal networks, which also means that the same parameters lead to the same result. However, with BNFinder one can use a number of input arguments such as scoring functions

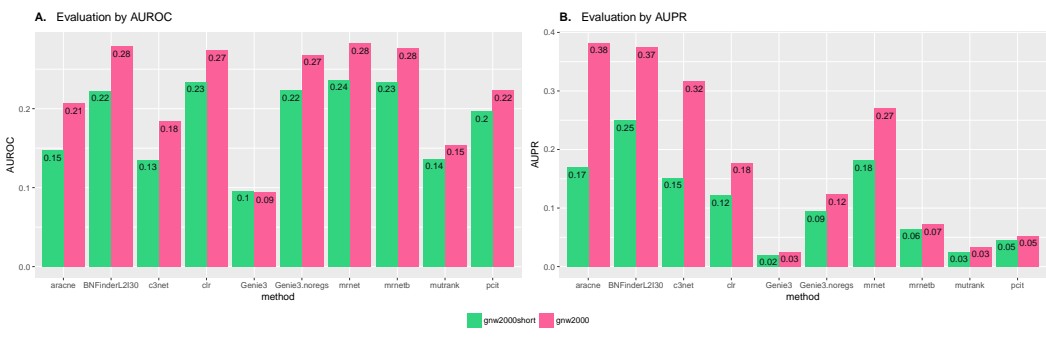

**Figure 10   GNW 2000 genes Yeast synthetic dataset, evaluation by NetBenchmark package.** Area under ROC (A) and PR (B) curves values are considered. GNW 2,000 short dataset contains only 150 observations, while GNW 2,000 has all the 2,000 observations. The effect of observations number increase is clearly seen for all the methods. Genie3.noregs is the result of Genie3 execution without the regulators list. BNFinder is used with parents sets limit 2 and suboptimal parents sets 30.

**Table 5   The number of the interactions in the output network based on different BNFinder parameters.** DREAM2 Genome Scale Network Inference data is used.

|                  | Subparents = 25 | Subparents = 50 | Subparents = 100 |
| ---------------- | --------------- | --------------- | ---------------- |
| Parent limit = 1 | 78,366          | 156,685         | 313,061          |
| Parent limit = 2 | 65,108          | 116,424         | 213,389          |

(Bayesian-Dirichlet equivalence, Minimal Description Length or Mutual information test), static or dynamic (also with self-regulatory loops) BNs, perturbation data, or even prior information on the network structure. All of these may alter results significantly, so, naturally we are interested in choosing best parameters for a particular dataset. Here we studied the impact of two very important parameters: parents sets limit and number of suboptimal parents sets (gives alternative sets of regulators with lower scores).

In Table 5 we have summarized the total number of interactions returned by BNFinder with different maximal parents per gene and different number of suboptimal parent sets. The results indicate that increasing the size of the allowed parent set leads to the decrease in the total returned edges in the network. This may seem surprising at first, but it is consistent with highly overlapping suboptimal parents sets. Theoretically, increasing parents set limit should lead to better precision, while increasing the number of suboptimal parents set should increase the number of false positives by adding lower scored parents. However, it may depend on the particular dataset, especially on the number of observations available. On top of that, in cases where using higher number of parents per variable is computationally challenging, suboptimal parents may compensate for this limitation.

We studied the effect of different parameters by plotting AUPR against AUROC values. Figure 11 shows that on 2,000 genes × 2,000 experiments synthetic dataset using two parents per gene is always better than one, and increasing the number of suboptimal parents leads to an increase in AUROC and a slight decrease in AUPR values. Inferring a network from the same dataset with only 150 experiments sometimes resulted in a lower
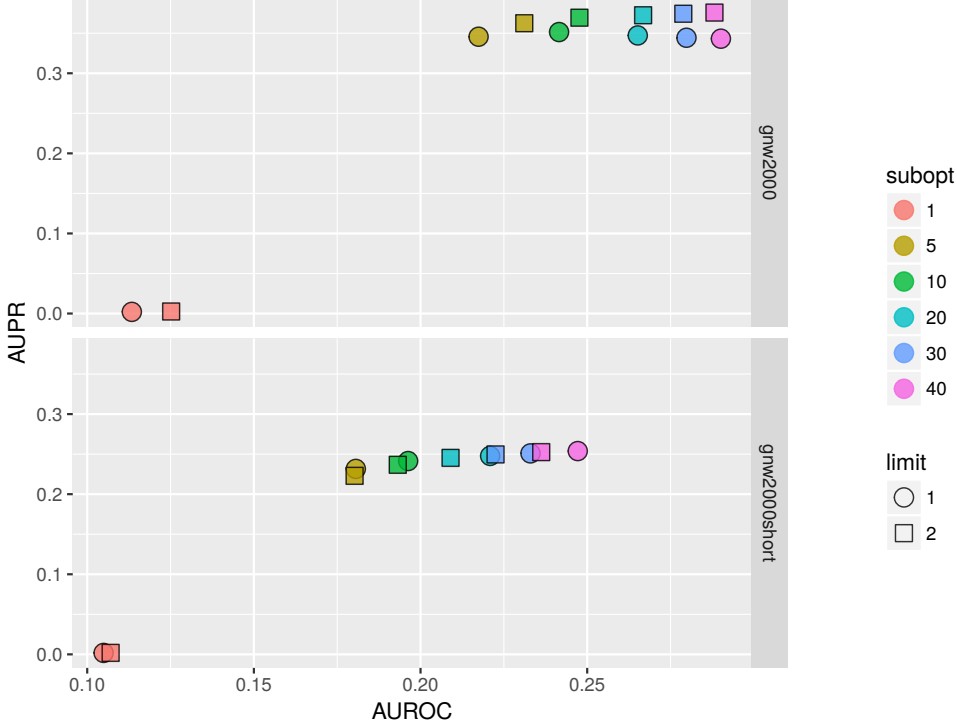

**Figure 11** **BNFinder parameter space examination on GNW 2000 genes Yeast synthetic dataset, evaluation by NetBenchmark package.** GNW 2,000 short dataset contains only 150 observations, while GNW 2000 has all the 2,000 observations. *Limit* stands for parents set size limit, while *subopt* denotes the number of suboptimal parents set in the output network.

AUPR for two parents per gene cases in comparison with singleton parents sets. While BNFinder scoring function penalizes for parents set size increase it might still produce false positives when the number of observational points is low and the number of variables is more than ten folds bigger. Importantly, zero (or very low) number of sub-optimal parent sets leads to extremely sparse output network (the number of edges is much less than 20% of all possible interaction) and therefore poor AUPR and AUROC values.

In general, we can conclude that if the user is interested in the very top of the strongest interactions in the network, one should use small numbers of sub-optimal parent (up to 5) sets and small limit on the parent-set size (up to 3). However, if one is interested in discovering the more global picture of the true regulatory network, one should focus on the higher number of sub-optimal parent sets with limit on the set size as high as it is computationally feasible.

The results of all performance and accuracy tests with the scripts to generate all the plots are available from a dedicated github repository—https://github.com/sysbio-vo/article-bnf-suppl.

## DISCUSSION

### Performance

Despite seemingly complex behavior of the **hybrid** algorithm and many cases where the **hybrid** and the **set-wise** algorithms can be applied, we can give users the best practice guidance for BNFinder application. In case of small networks where number of variables is 2 or more times less than the number of cores it is advised to use the **hybrid** algorithm, because the **set-wise** would generate more context switching events per variable. The same applies when a user imposes parent set limit equal or less than 3, which makes the computational load per variable more even. In case when the complex layered structure of regulators is introduced it is always better to use the **set-wise**. In case of big networks, when the number of variables is greater than the number of cores in a single computational unit, one can also use the new BNFinder feature, which allows to define subset of genes as it was described in **Distributed computations testing** section of this paper. Defined subsets can be computed separately and simultaneously on different computational units, and user should apply the same logic when choosing parallelization algorithm as for the smaller networks.

And finally, the user may just use the default parameters as **set-wise** algorithm did not show major drop in performance in comparison to the **hybrid** one.

### Accuracy of reconstruction

While we understand that there are many more tools for gene regulatory networks reconstruction in the literature we believe that NetBenchmark package is representative for the field since it incorporates state-of-the-art methods, which are based on a variety of different algorithms. On top of that using benchmarking tool makes it easier for other researchers to compare their methods to our results.

Measuring AUROC and AUPR values on 14 different datasets revealed that studied methods behave differently on different datasets, and none of the methods scored best in all cases. In general, time series data proved to be more challenging for the methods than inferring network from static gene expression datasets. Our results on 10 genes networks evaluation with top 20% and 100% interactions showed that such small networks can hardly be used as the only source of comparison.

Testing BNFinder on the mentioned datasets we concluded that it performed best on the static gene expression datasets with additional prior knowledge (transcription factors list, prior edge probability between genes), while for other methods such as Genie3 the same information did not yield significant improvement.

It should be noted, that in most of these scenarios, our knowledge of the true network connections is also limited and different tools use different definitions of a meaningful "connection", therefore it is expected that, for example, a tool using mutual information will capture a different subset of connections than a tool using Bayesian-Dirichled equivalence. While our opinion is that such automated methods will be gaining importance as their accuracy increases, there is still a lot of work ahead of us on careful validation of the problems various tools have and on refinement of the definitions of regulatory interactions.

## CONCLUSIONS

The improvement over previous version of BNFinder made it feasible to analyze datasets that were impossible to analyze before by utilizing the power of distributed and parallel computing. It allowed us to significantly extend the application range of the tool, and, for the first time, to compare it with the best-performing non-Bayesian methods. BNFinder showed overall comparable performance on synthetic and real biological data, providing significant advantage in cases when prior knowledge on genes interactions can be introduced. This can lead to further research on the optimization of the BNFinder method for the purpose of finding larger networks with better accuracy. We provide the new BNFinder implementation freely for all interested researchers under a GNU GPL 2.0 license.

## ACKNOWLEDGEMENTS

For distributed calculations authors used computational clusters of Taras Shevchenko National University of Kyiv, Institute of Molecular Biology and Genetics of NASU, Institute of Food Biotechnology and Genomics of NASU, joint cluster of SSI "Institute for Single Crystals" and Institute for Scintillation Materials of NASU. We would like to express out gratitude to Prof. Maria Obolenska from Institute of Molecular Biology and Genetics, Kyiv, Ukraine for the proofreading and critique. We also thank Dr. Veronika Gurianova from Bogomoletz Institute of Physiology, Kyiv, Ukraine for her valuable comments on the article visuals, which helped us to make the plots more clear and understandable. We are extremely grateful to Dr. Pau Bellot from Centre for Research in Agricultural Genomics (CRAG), Barcelona, Spain, one of the NetBenchmark R package authors and maintainers, for the assistance with package related problems. Together we were able to fix two bugs, which under specific circumstances might have hindered evaluation results. The virtue of Open Source has yet again proven to positively influence the scientific advance.

### Funding

This work was partially supported by the National Center for Science grant (decision number DEC-2012/05/B/N22/0567) and Foundation for Polish Science within the SKILLS programme. This work was also supported by National program of Grid technologies implementation and usage in Ukraine (project number 69-53/13 and 0117U002812). There was no additional external funding received for this study. The funders had no role in study design, data collection and analysis, decision to publish, or preparation of the manuscript.

### Grant Disclosures

The following grant information was disclosed by the authors:
National Center for Science grant: DEC-2012/05/B/N22/0567.
SKILLS programme.
National program of Grid technologies implementation and usage in Ukraine: 69-53/13, 0117U002812.

## Competing Interests

The authors declare there are no competing interests.

## Author Contributions

- Alina Frolova conceived and designed the experiments, performed the experiments, analyzed the data, prepared figures and/or tables, authored or reviewed drafts of the paper, approved the final draft.
- Bartek Wilczyński conceived and designed the experiments, authored or reviewed drafts of the paper, approved the final draft.

## Data Availability

BNFinder tool source code:

https://github.com/sysbio-vo/bnfinder.

Scripts to generate figures, run methods etc:

https://github.com/sysbio-vo/article-bnf-suppl.

## Supplemental Information

Supplemental information for this article can be found online at http://dx.doi.org/10.7717/peerj.5692#supplemental-information.

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
