# Peer review of "Distributed Bayesian networks reconstruction on the whole genome scale"

_PeerJ, doi:10.7717/peerj.5692_

## Round 0.1 · original submission · Minor Revisions

Both reviewers raise valid concerns regarding the structuring of the manuscript and presentation of the findings. The description of your tool and its utility require substantial more details to make it better accessible and user-friendly to biologists, and hence enhance the chance of it being implemented by prospective users. Please carefully address the issues raised by the two reviewers.

Reviewer 1 ·

Basic reporting

Bayesian networks are useful models for inferring regulatory interactions in biological systems. The authors present an improved version of BNFinder (Wilczyński and Dojer, 2009), a tool to learn Bayesian networks in polynomial time (as opposed to being an NP-hard problem) if external constrains are applied. Specifically, previous implementations of parallelisation were vulnerable in cases where the node degree is highly variable, leading to uneven thread load (Dojer et al., 2013). This is common is biological networks, which typically exhibit scale-free topologies, meaning that the performance gain in the previous version of the tool was limited. Thus, the application of BNFinder on genome-wide datasets was not feasible. In this paper, the authors present a more sophisticated parallelisation algorithm, which they claim dramatically increases computational efficiency and will allow Bayesian network construction on a genome-wide scale.

However, I have several comments and concerns which I believe should be addressed by the authors before publication. In general, I found the paper was poorly structured, there was not a sufficient explanation of how the tool works, and the paper did not sufficiently explain how this tool could be useful to biologists. I believe that this paper is only interpretable to someone who already has a comprehensive understanding of the tool, and that new users will struggle to understand the tool (as was the case for me) unless significant improvements are made to the paper.

Most critically, I found it difficult to assess the tool’s utility because there was not sufficient explanation of how it works nor what data is required to run the tool. For example, I particularly struggled with what external constraints must be placed in order to ensure the acyclicity of the graph. As I could understand from the paper and the online tutorials provided for BNFinder, an expected layout structure must be provided for the network (lines 108-113), with the authors also stating that BNFinder is particularly suited to cases where prior knowledge on the network is provided (line 323). It would be greatly appreciated if the authors could elaborate on what kind of input is required by BNFinder, and how BNFinder works differently depending on the different types of inputs (this is briefly mentioned in lines 262-265, but much more description is required, and should appear earlier in the paper). In addition, if it is true that an expected layout is required, this should be explicitly stated by the authors, as well as explaining how they derived the expected layout for the Sachs et al., 2005 dataset (lines 108-113), in addition to the expected layouts for all other networks constructed. Furthermore, in the case that this tool is used for genome-wide network construction, I struggle to understand what the expected layout would look like (other than a network itself), which would suggest that this tool cannot be used in de novo network construction. Again, if this is the case, this should be explicitly stated by the authors. Furthermore, the authors claim that their method for constructing Bayesian networks is superior to heuristic search methods as they can find the optimal network. However, if the tool is only capable of finding the optimal solution for a given set of constraints in order to ensure acyclicity, then this is not the optimal network either and the comparison is unfair.

On a similar point, I believe that non-computational biologists will have particular difficulty in understanding how this tool works and how to apply it to their data. Given that –omics datasets are becoming more commonplace in labs without a strong computational background, there is a growing need for tools which can analyse these datasets without the need to be a specialist. As the authors acknowledge (line 34), Bayesian networks are ideal for this, as the networks are concise and easily interpretable. Thus, it would greatly increase the utility of the tool and the readership of this paper if part of the introduction were targeted towards non-computational biologists.

I found that the paper was littered with minor grammatical errors. In general, the paper needs to be more thoroughly checked for grammatical correctness. For example, the following corrections are suggested (although there are many other mistakes):
Line 69: It is possible to consider different possible parent sets in parallel, as is the case for the set-wise algorithm.
Line 72: The set-wise algorithm,
Line 73: in comparison with the variable-wise.
Line 101: There is no regulators list for this network, so BNFinder constructed a dynamic Bayesian network.
Line 262: with BNFinder one can use a number of inputs

Experimental design

I had some difficulties in understanding the differences in the parallelisation algorithms employed. Firstly, I did not find the python pseudocode (Figures 1-3) helpful, as this still required knowledge of what each variable is and the custom functions used by the algorithm. I would suggest flow diagrams as a more appropriate way to convey this information, although in either case, it is not appropriate to simply give the names of custom functions used by the tool without providing any guidance as to what they do. Furthermore, it is not explained how the set-wise algorithm was capable of grouping genes into sets (singleton parents, two-set parents etc., lines 69-77), which seems to be the critical step in efficient parallelisation. Further clarification on this is required.

I appreciated the extensive tests and benchmarking exercises which were conducted by the authors. Furthermore, they were very open about the difficulties in using benchmarking exercises to evaluate algorithm performance, and that BNFinder only performed best for some datasets. This is rare in papers describing novel computational tools, and it will prove useful to potential users in deciding which tool to use for their data. It would be particularly helpful if it is stated in the abstract that BNFinder is particularly suited to constructing networks from static gene expression data (lines 246-247).

Validity of the findings

I also have some concerns about the conclusions derived from the performance testing of the new algorithm. In particular, the authors show in Figures 4 and 5 that the performance of their new algorithm is better than the previous version. However, this is only tested on datasets with 20 and 11 genes respectively, so does not support their conclusion that their new algorithm will facilitate genome-wide network construction. It would improve the scientific robustness of the paper if comparison of parallelisation algorithms were performed on appropriately sized datasets. Furthermore, I question the authors assertion that their novel algorithm will facilitate genome-wide network construction. According the Table 1, it took over 12 years of CPU time in order to construct a network a network with ~3500 genes, with each gene limited to 3 parents. Whilst lowering the maximum number of parent genes means that the network could be constructed in a more reasonable time (although still requiring high-performance computing), I am concerned that limiting the number of parents will result in networks which do not exhibit scale-free topology, and thus whether the networks provided by BNFinder will recapitulate the real network structure. I would be interested to hear the authors opinions on this issue.

Reviewer 2 ·

Basic reporting

This paper considers new parallelised algorithms designed for multi-core and distributed systems for the problem of Bayesian Network Reconstruction. The authors have worked in the past on an algorithm and associated software tool (BNfinder) for this. Here, they explore how to scale this algorithm up to much larger networks. They have implemented and added their new schemes to the BNFinder and they have tested this on several simulated and real datasets. Overall, the paper has very good basic reporting, there is sufficient explanation of the background and context. Also the results are clearly and professionally presented.

- One area that presentation can improve is the captions of the figures. Currently, they are very short and do not provide sufficient information on what is shown at times. For example, speedup and efficiency need to be described clearly in the caption of Figure 4. How are the methods ranked in Figures 7-10?

Experimental design

This article develops new computational statistical methodology with direct application to global biological data such as transcriptomics as illustrated in the application of the tool on read data. There is a clear aim here regarding new schemes and this is regorously investigated. I have the following comment to improve the presentation of experimental design.

- Is the idea of hybrid method novel, if so clearly state this and if not please cite a relevant paper. As they argue by the equation on page 15, it is theoretically expected the hybrid method to do worse than the set-wise. Could they explain why in practice hybrid method could have advantage in certain situations.

Validity of the findings

This is a computational project and difficult to assess directly the validity of the results. But, the approach and methodology used is sound and claims are well supported by the results. A bit longer discussion of the comparisons with other methods would be welcome.

---

## Round 0.2 · accepted · Accept

The authors have thoroughly addressed the issues raised by the reviewers.

#